# The Double Mediating Effect of Social Isolation and Emotional Support on Feelings of Entrapment and Motivation for Recovery among Korean Alcoholic Inpatients

**DOI:** 10.3390/ijerph18094710

**Published:** 2021-04-28

**Authors:** Joo-Young Lee, Jae-Sun An, Kyung-Hyun Suh

**Affiliations:** 1Department of Addiction Science, Sahmyook University, Seoul 01795, Korea; shinic78@naver.com; 2Department of Counseling Psychology, Sahmyook University, Seoul 01795, Korea; lucky603@hanmail.net

**Keywords:** alcoholics, entrapment, isolation, support, motivation, recovery

## Abstract

This study identified the relationship between feeling of entrapment and motivation for change among hospitalized alcoholic patients and examined the double mediating effect model of social isolation and emotional support on this relationship. The study participants were 101 male and female alcoholic patients hospitalized at C hospital, which specializes in alcohol treatment at I city in Korea. PROCESS Macro 3.5 Model 6 was used for analyses of double mediating effects. The results revealed that entrapment and social isolation were negatively correlated with motivation for recovery of alcoholic inpatients, whereas emotional support was positively correlated with it. In a sequential double mediation model for motivation to change in alcoholic inpatients, the direct effects of social isolation and entrapment were not significant. However, the sequential indirect effect of social isolation and emotional support on entrapment and motivation for recovery among alcoholic inpatients was significant. These results suggest that making alcoholic inpatients not feel socially isolated by providing them with emotional support or through other means of assistance by practitioners or family members is important for their recovery from alcohol use disorder.

## 1. Introduction

There exists a permissive atmosphere for drinking in Korean society, and generous social norms for drinking, even for binge drinking, exposing Koreans to problematic drinking [1,2,3]. For this reason, the prevalence of alcoholism, which is diagnosed as an alcohol use disorder, and the drinking rate is maintained at a high level in Korean society [4]. Over the past decade, middle-aged men and young women have had a high drinking rate in Korea, and both Korean men and women were often treated for alcohol use disorders in their 40s and 50s. As a result, the most frequent reason for Koreans’ hospitalization in their 50s was alcohol use disorder [5]. Since Korea’s birth rate is globally the lowest [6], it has become more important than ever to treat Korean alcoholic patients. Therefore, in this study, we aim to investigate psychosocial variables that could predict the motivation for change or recovery of Korean alcoholic patients.

First, we choose entrapment as a variable. This is because people who are addicted feel trapped by addiction agents [7], and alcoholic patients might feel even more entrapped when hospitalized. A longitudinal study has also proven that entrapment makes people depressed and anxious [8]; consequently, it makes people give up on their lives [9]. Such a relationship might also be more prominent in people who are institutionalized [10]. The feelings of entrapment could reduce the motivation for life and may demotivate people for changing [11]. Therefore, in this study, it is assumed that feelings of entrapment negatively affects the motivation for recovery among hospitalized patients with alcoholism.

Nevertheless, we did not assume that entrapment would directly affect the motivation of hospitalized alcoholic patients to change their drinking behaviors. We believed that there are some mediators between feelings of entrapment and their motivation for recovery. Further, it was assumed that social isolation mediates the relationship between entrapment and motivation for recovery among alcoholic patients. Researchers have long been interested in feelings of loneliness or social isolation related to alcohol problems [12]. The feeling of social isolation could also result in some form of addiction. A study found that social isolation makes middle-aged and old-aged Korean adults addicted to smartphones [13]. Brain research has also proven that social isolation makes people more attracted to alcohol use and resulting in the continuation to drink, even if it is problematic [14]. Therefore, this study assumes that the feelings of entrapment of alcoholic inpatients could make them feel socially isolated; consequently, their motivation for recovery too would be reduced. 

In this study, it is assumed that although social isolation may reduce the motivation for recovery among alcoholic patients directly, the feelings of entrapment and isolation experienced by alcoholic inpatients also leads to less emotional support, resulting in even lesser motivation for recovery. This is because social isolation is closely related to the lack of emotional support [15], and less emotional support leads to less motivation for recovery [16]. Some studies found that social support, including emotional support, plays an important role in motivating alcoholic patients to change their problematic drinking behaviors [17,18]. Possibly, there is also less emotional support if inpatients are entrapped; therefore, this study tried to verify a model that include a path from a sense of entrapment to the motivation for recovery through emotional support. Moreover, the model includes a sequential double mediation effect of social isolation and emotional support between feelings of entrapment and motivation for recovery of alcoholic inpatients (Figure 1).

## 2. Methods

### 2.1. Participants

The participants were 101 male and female alcoholic patients diagnosed with alcohol use disorder and hospitalized in Chamsarang hospital, which specializes in alcohol treatment at Incheon, Korea. The patients who participated in the study were hospitalized in Wards 3, 4, and 5. Ward 3 was for patients with only alcohol use disorder, Ward 4 was for male patients with alcohol use disorder and mental disorders as co-occurring disorders (COD), and Ward 5 was for female patients with both alcohol use disorder and mental disorders. 

Inclusion criteria are as follows. First, participants were inpatients aged 18 or older and who were diagnosed with alcohol use disorder based on the DSM-5 (Diagnostic and Statistical Manual of Mental Disorders, 5th edition) criteria [19]. Second, participants were alcoholic patients who were able to communicate, understand, and respond to items in the questionnaires. To meet the second criterion, alcoholic patients with intellectual disabilities such as mild cognitive impairment and dementia, and schizophrenia spectrum disorders such delusional disorder, schizophrenia, and schizoaffective disorder as a COD were excluded. The survey was conducted on 130 patients, but 10 patients gave up while filling the questionnaire, and 19 of the 120 questionnaires were excluded because they were incomplete. Finally, questionnaires from 101 participants were included in the analysis.

### 2.2. Measures

#### 2.2.1. Feelings of Entrapment

The participants’ entrapment stress was measured with the Korean version of the Entrapment Scale (K-ES) [20,21]. K-ES consists of 16 items and two sub-factors: Internal entrapment (eight items) and external entrapment (six items). However, only the entrapment total score was used in the analysis for this study. Each item was rated on a five-point Likert scale ranging from 1 (not at all true) to 5 (very true). The internal consistency of the 16 items (Cronbach’s α) was 0.95 in this study.

#### 2.2.2. Social Isolation

Social isolation was measured with modified social isolation subscale items from Lee’s Alienation Scale [22,23]. This scale consists of 28 items and 4 sub-factors: Helplessness (7 items), meaninglessness (7 items), loss of social norm (7 items), and social isolation (7 items). Only the social isolation subscale was used for this study. Items were slightly modified for patients’ situations. Each item was rated on a four-point Likert scale ranging from 1 (not at all true) to 4 (very true). The internal consistency of the seven items (Cronbach’s α) was 0.90 in this study.

#### 2.2.3. Emotional Support

To measure the emotional support that participants received, we used the emotional support subscale from Park’s social support scale [24]. This scale measures four types of perceived social supports from others: Emotional support (ten items), informational support (five items), material support (five items), and appraisal support (five items). Only the emotional support subscale was used for this study. Each item was rated on a five-point Likert scale ranging from 1 (not at all true) to 5 (very true). The internal consistency of the items (Cronbach’s α) was 0.95 in this study.

#### 2.2.4. Motivation for Recovery

Participant’s level of motivation for recovery was measured using the Korean version of The Stages of Change Readiness and Treatment Eagerness Scale (K-SOCRATES) [25,26]. K-SOCRATES consists of 19 items, and measures three aspects of motivation for changes in alcoholics: Recognition (8 items), ambivalence (4 items), and taking a step (7 items). Similar to Yoon and Kim’s study [27], we measured alcoholics’ motivation for recovery with the subscale of “taking a step”. One item was excluded because it is a treatment-related question that inpatients may have left uncomfortable responding to while in hospital. Only six items were used in this study. This scale uses a five-point Likert scale ranging from 1 (not at all true) to 5 (very true). In this study, internal consistency (Cronbach’s α) was 0.94 for the six items. 

### 2.3. Procedure

Before conducting this study, we were approved by the Institutional Review Board (approval number: 2-1040781-AB-N-01-2017017HR), and all research processes were conducted ethically. For this study, data were collected with informed consent from patients.

### 2.4. Statistical Analysis

The data were analyzed with IBM SPSS Statistics for Windows 23.0, and PROCESS Macro 3.5 was used for this study. Skewness and kurtosis of the data were checked for parametric statistical analyses. Pearson’s product moment correlational analysis was conducted using SPSS, and analysis of a sequential double moderating mediating effect was performed with PROCESS Macro 3.5 Model 6 [28]. Finally, bootstrapping using 5000 resamples with 95% confidence interval was used to analyze the mediating model’s significance.

## 3. Results

### 3.1. Characteristics of Patients who Participated in this Study

Among the participants, 94 were male (93.1%) and seven were female (6.9%). The average age of the participants was 52.74 ± 8.48 years. The first time they used alcohol was most often reported as from the age of 10 and upwards and, that they started using alcohol frequently while in their 20s (Table 1).

Among the inpatients who participated in the study, the ones most frequently diagnosed with alcohol use disorder were in their 40 s. The average length of hospitalization reported by participants was 195.35 ± 123.91 days, with six people hospitalized throughout the year. About half of the patients were hospitalized for more than six months. Among them, 30 patients (29.7%) had COD.

### 3.2. Relationship between Variables Involved in Motivation for Recovery of Alcoholic Patients

Table 2 presents the correlational analysis of feelings of entrapment, social isolation, emotional support, and motivation for recovery among Korean alcoholic patients. None of the absolute values for skewness and kurtosis exceeded 2, indicating that all variables’ variances were close to the normal distribution for conducting parametric statistical analyses [29].

The correlational analysis revealed that entrapment (*r* = −0.322, *p* < 0.01) and social isolation (*r* = −0.260, *p* < 0.01) negatively correlated to motivation for the recovery among alcoholic patients, while emotional support positively correlated to motivation for recovery (*r* = 0.526, *p* < 0.001). Entrapment was positively correlated to social isolation (*r* = 0.713, *p* < 0.001). These two variables shared almost 51% of the variation. Moreover, entrapment (*r* = −0.482, *p* < 0.001) and social isolation (*r* = −0.540, *p* < 0.001) negatively correlated to emotional support for alcoholic patients.

### 3.3. Verification of the Double Mediation Model for the Motivation for Recovery among Alcoholic Patients

This study examined a mediating effect of social isolation, emotional support on feelings of entrapment, and motivation for recovery among Korean alcoholic patients (Table 3). It is known that a statistical multicollinearity problems occur when tolerance is less than 0.2 or 0.1, and variance inflation factors (VIF) are greater than 5 or 10 [30]. Since tolerance of predictors in this study were 0.442~0.689, and VIFs were 1.451~2.265, the multicollinearity problem was not significant. Additionally, a value of the Durbin Watson statistic was 2.251, which indicates that there is no autocorrelation detected in the sample as it was close to 2.

The result shows that entrapment positively influenced social isolation (*β* = 0.229, *p* < 0.001), but it did not significantly directly influence emotional support (*β* = −0.118, *p* = 0.103) and motivation for the recovery (*β* = −0.070, *p* = 0.157) among alcoholic patients in this model. Moreover, social isolation negatively influenced emotional support (*β* = −0.742, *p* < 0.01) for alcoholic patients, but it did not significantly or directly influence motivation for recovery (*β* = 0.180, *p* = 0.256). In addition, emotional support positively influenced motivation for the recovery among alcoholic patients (*β* = 0.344, *p* < 0.001).

Figure 2 shows that feelings of entrapment significantly influenced motivation for the recovery among alcoholic patients (*β* = 0.229, *p* < 0.001), but it became insignificant when social isolation and emotional support were added. This means that social isolation and emotional support can completely sequentially mediate feelings of entrapment and motivation for the recovery among alcoholic patients in this model.

Using 95% bootstrap confidence intervals from 5000 bootstrap replications, the double mediating effect of social isolation and emotional support in the relationship between entrapment and motivation for recovery among alcoholic patients was verified, and the results are presented in Table 4.

In this model, the total mediating effect was −0.058 (−0.1534~0.0431), which was proven to be non-significant because no relationship exists between the upper and lower bounds of bootstrapping at 95% confidence intervals. Verifying the simple mediating effect revealed that the path from entrapment to motivation for recovery via social isolation was not significant (−0.0317~0.1250). Further, the path from entrapment to motivation for recovery via emotional support was also not significant (−0.1045~0.0182). However, the sequential double mediating effect of social isolation and emotional support on entrapment and motivation for recovery (entrapment → social isolation → emotional support → motivation for recovery) was 0.012 (−0.1135~−0.0172), which was significant.

## 4. Discussion

The present study explored the relationships between the feeling of entrapment, social isolation, emotional support, and motivation for recovery among hospitalized alcoholic patients in Korea. Further, it examined the double mediating effect of social isolation and emotional support on feelings of entrapment and motivation for the recovery among alcoholic inpatients. These attempts have produced valuable information for further studies as well as for professionals who help alcoholic patients recover from alcohol use disorder, and the implications are discussed below.

First, entrapment and social isolation among alcoholic inpatients were closely correlated. Entrapment shared a 50.8% variance (*r* = 0.713) with the social isolation of alcoholic inpatients. If a patient is admitted to the hospital, he or she feels trapped and isolated because they are quarantined, while also undergoing contact precautions [31]. In particular, patients within psychiatric inpatient care are more likely to feel social isolation; the treatment of this feeling may encourage and promote recovery from psychiatric disorders [32]. Considering this study’s results, it was concluded that patients hospitalized with alcohol use disorder had no choice but to feel entrapped and experience social isolation, as proven by previous studies as well [31,32].

As hypothesized in this study, the more the feelings of entrapment and social isolation experienced by alcoholic inpatients, the less emotional support they felt. Notably, patients trapped in a hospital and socially isolated due to alcohol use disorder find it difficult to receive emotional support from others in their environment. A previous study found that inpatients were easily or highly emotionally distressed. Further, many patients need emotional support from medical professionals such as physicians and nurses rather than from psychologists or pastors [33]. These results indicated that it is important to provide emotional support to hospitalized patients with alcohol use disorder. Indeed, providing emotional support has been considered an important factor in the recovery program for patients with substance use disorder, including alcohol use disorder [33,34].

As hypothesized in this study, feelings of entrapment, social isolation, and receiving emotional support were significantly correlated with motivation for recovery among alcoholic inpatients. This means that the more alcoholic inpatients feel entrapped, experience social isolation, or the lesser emotional support they receive, the less motivated they are to change their drinking behaviors. However, the sequential double mediation model verified in this study revealed that social isolation and feelings of entrapment only influence the motivation for recovery among alcoholic inpatients through emotional support. In other words, if alcoholic inpatients are feeling entrapped and socially isolated, they receive less emotional support, which lowers their motivation for recovery. Therefore, it is not only family members but also medical personnel who need to provide more emotional support to hospitalized patients with alcohol use disorder. After qualitatively analyzing the emotional support received during treatment for recovery from severe alcohol use disorder, Brook et al. concluded that it must be “operationalized as the moral support (p. 70)” specific to be recovered from alcoholism. This means that providing emotional support to alcoholic patients is needed to reduce the stigma effect caused by a moral model-based approach to addiction. Therefore, health professionals should understand patients’ feelings and needs, and provide encouragement [35]. In conclusion, this study reiterates that emotional support for alcoholic patients is a determinant for their motivation to change drinking behaviors, which is a key factor in the treatment of patients with severe alcohol use disorder.

This study found that social isolation and emotional support sequentially mediated feelings of entrapment and motivation for recovery among Korean alcoholic inpatients. However, there are some limitations to be considered. First, the sample of this study is not representative of all Korean patients hospitalized with alcohol use disorder because the sample size was not large, and the proportion of male patients was too high, and data were collected from a hospital located only in a certain area of Korea. The patients with severe symptoms could not answer psychological tests, and some patients were not motivated to answer all items in the questionnaires. Therefore, further studies are needed to re-verify the results of this study. Second, although we assumed the cause-and-effect relationship based on previous studies’ results and logical sense, the cause-and-effect relationship cannot be completely concluded based on the results of a correlational study. Despite these limitations, this study’s results could academically contribute to further studies and clinically contribute toward planning the treatment for patients with alcohol use disorder.

## Figures and Tables

**Figure 1 ijerph-18-04710-f001:**
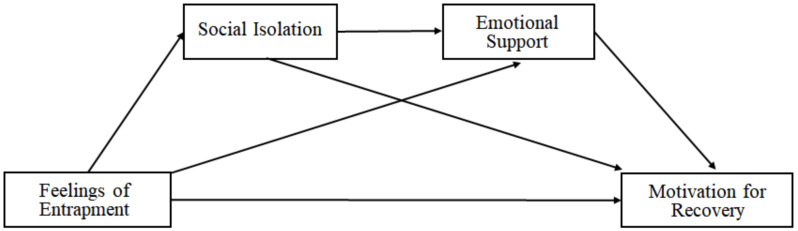
Double sequential mediation model.

**Figure 2 ijerph-18-04710-f002:**
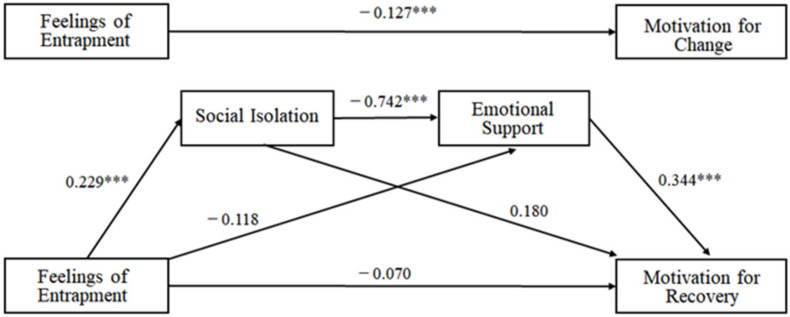
Examined double mediation model of social isolation and emotional support on feelings of entrapment and motivation for recovery of alcoholic patients (*** *p* < 0.001).

**Table 1 ijerph-18-04710-t001:** Characteristics of participants (*N* = 101).

Variables		Frequency	Percent (%)
Age of first alcohol use	10s 20s 30s or above Unanswered	66 31 3 1	65.3 30.7 3.0 1.0
Age at which regular drinking began	10s 20s 30s or above Unanswered	22 56 20 3	21.8 55.4 19.8 3.0
Age at which they were diagnosed with alcohol use disorder	30s or below 40s 50s or above Unanswered	27 37 24 13	26.7 36.6 23.8 12.9
Hospitalization periods for the past year	Below 3 months 3~6 months Over 6 months Unanswered	21 16 47 17	20.8 15.9 46.5 16.8
Presence of COD	Having COD None	30 71	29.7 70.3

**Table 2 ijerph-18-04710-t002:** Correlational matrix of entrapment, social isolation, emotional support, and motivation for recovery of alcoholic patients (*N* = 101).

Variables	1	2	3	4
1. Feelings of entrapment	1			
2. Social isolation	0.713 ***	1		
3. Emotional support	−0.482 ***	−0.540 ***	1	
4. Motivation for recovery	−0.322 **	−0.260 **	0.526 ***	1
*M*	51.47	17.82	27.43	17.63
*SD*	16.93	5.45	10.11	6.68
Skewness	−0.29	0.07	0.21	0.14
Kurtosis	−1.10	−0.77	−0.83	0.96

** *p* < 0.01, *** *p* < 0.001.

**Table 3 ijerph-18-04710-t003:** Double mediating effect of social isolation and emotional support on feelings of entrapment and motivation for recovery of alcoholic patients.

Variables		*β*	*S.E.*	*t*	LLCI	ULCI
	Mediating Variable Model (Outcome Variable: Social Isolation)					
Constant		6.026	1.227	4.91 ***	3.5904	8.4613
Feelings of entrapment		0.229	0.023	10.11 ***	0.1842	0.2742
	Mediating Variable Model (Outcome Variable: Emotional Support)					
Constant		46.693	3.022	15.45 ***	40.6946	52.6905
Social isolation		−0.742	0.222	−3.34 **	−1.1820	−0.3012
Entrapment		−0.118	0.071	−1.65	−0.2592	0.0241
Dependent Variable Model (Outcome Variable: Motivation for Recovery)						
Constant		8.554	3.773	2.27 *	1.0652	16.0420
Social isolation		0.180	0.158	1.14	−0.1329	0.4933
Emotional support		0.344	0.068	5.06 ***	0.2090	0.4791
Feelings of entrapment		−0.070	0.049	−1.43	−0.1662	0.0272

* *p* < 0.05, ** *p* < 0.01, *** *p* < 0.001. Note. LLCI: lower level for confidence interval; ULCI: upper level for confidence interval.

**Table 4 ijerph-18-04710-t004:** Indirect effects of the mediation model.

Path	Effect	*S.E*	BC 95% CI
Entrapment → Social isolation → Motivation for recovery	0.041	0.039	−0.0317~0.1250
Entrapment → Emotional Support → Motivation for recovery	−0.040	0.031	−0.1045~0.0182
Entrapment → Social isolation → Emotional support → Motivation for recovery	−0.059	0.025	−0.1135~−0.0172
Total indirect effect	−0.058	0.049	−0.1534~0.0431

## Data Availability

Any queries regarding the data used in this study may be directed to the corresponding author. The dataset used in the present study is available on reasonable request.

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
