# Peer review of "The Double Mediating Effect of Social Isolation and Emotional Support on Feelings of Entrapment and Motivation for Recovery among Korean Alcoholic Inpatients"

_ijerph, 2021, doi:10.3390/ijerph18094710_

Round 1

Reviewer 1 Report

Dear authors,

Please find my feedback in the attached document. My main feedback is that the authors try to shorten the paper further by avoiding repetition of points in the Discussion, repetition of avoidable phrases (e.g., "there was zero (-.0317 ~ .1250) between the upper and lower levels for a confidence interval"), and improvement in presentation of tables. There are also some issues with the language so please improve that.

Author Response

Thank you very much for your comments to improve the quality of this articles.

The revised parts were marked in red, we included the page and line of the revised part.

Response to Reviewer 1 Comments

Point 1. Typo and stigmatizing sentence

Response 1: To my shame, there were typos and wrong word choices, I corrected it. There was still a problem even though we asked an English correction and editing company to correct it in English. I will request English correction one more time later.

Frequently → frequent (p. 1, L 23)

addicts → people who are addicted (p. 2, L 3)

alcoholics → alcoholic patients (p. 2, L 4)

alcoholics patients → alcoholic patients (p. 2, L 24)

while emotional support negatively → while emotional support positively (p. 6, L 7)

fell → feel (p. 9, L 10)

Point 2. Section needs improvement, currently it is very repetitive. It appears authors are making the same point by changing the words used/structure sentence etc.

Response 2: Thank you very much for your good comment on this part. We omitted the part “because zero exists between the upper and lower bounds of” you pointed out, so that they do not overlap (p. 8, L 7-11).

Point 3. A misleading sentence that was hard to understand “Hospitalization as a patient is bound to be subject to isolation precautions in the hospital setting, including contact precautions, which leads to feelings of entrapment and isolation. Patients are forced to experience a feeling of social isolation in psychiatric inpatient care” and “the lower the received emotional support”

Response 3: As you pointed out, the sentence was not clear. Therefore, we corrected it as follows. “If a patient is admitted to the hospital, he or she feels trapped and isolated because inpatient is being quarantined in the hospital setting, including contact precautions [31]. In particular, patients in psychiatric inpatient care are more likely to feel social isolation;” (p. 9, L 5-7) “the less the less emotional support they felt.” ” (p. 9, L 13).

Point 4. moral support??

Response 4: Thank you for your comment. As you pointed out, there was no detailed explanation of moral support. So, we corrected the sentence and added the explanation as follows. “operationalized as the moral support (p. 70)” specific to recover from alcoholism. This means that emotional support for alcoholic patients is needed to reduce stigma effect caused by a moral model-based approach to addiction. Therefore, ….” (p. 9, L 29-31).

Reviewer 2 Report

The study aim to identify the relationship between entrapment and motivation for change of hospitalized alcoholic patients and examine the mediating effect of social isolation and emotional support. 

Variables such as entrapment, social isolation and (lack of) emotional support are susceptible to overlap. Therefore colianerity and variance inflation factors must be reported. Failing to do this is a serious hazard to the validity of the results, and therefore any conclusions drawn from those.  

Author Response

Thank you very much for your comments to improve the quality of this articles.

The revised parts were marked in red, we included the page and line of the revised part.

Response to Reviewer 2 Comments

Point 1. Variables such as entrapment, social isolation and (lack of) emotional support are susceptible to overlap. Therefore colianerity and variance inflation factors must be reported. Failing to do this is a serious hazard to the validity of the results.

Response 1: Thank you for your good comment. As you pointed out, multicollinearity is that can significantly distort the results of a regression model based on correlation coefficients. So I checked the multicollinearity as you advised and found that it was not a problem. And the results were presented in the result section as follows.

“It is known that statistically multicollinearity problem occurs when a tolerance is less than .2 or .1, and/or a VIF is greater than 5 or 10 [30]. In this study, the multicollinearity problem was not significant because tolerance was .479 for entrapment, .442 for social isolation, and .689 for emotional support, and VIF was 2.089 for entrapment, 2.265 for social isolation, and 1.451 for emotional support.“(p. 6, L 14-18).

The concern about multicollinearity is probably because the correlation between entrapment and social isolation was too high (r=.713). But, you don't have to worry about, because entrapment and social isolation only share the variance, is not overlapping concepts.

If we look the items of questionnaires that measure each variable, we can understand. .

Example of items of entrapment: “I want to get away from my present self and start over.”, “I often have the feeling that I would just like to run away.”, “I want to run away from people around me who are more powerful than I am.”, and “I feel like I am trapping by others.”

Example of items of social isolation: “There is no one to complain when I am upset.” “I don't get much praise from other people.”, and “People around me are indifferent to me.”

Reviewer 3 Report

Dear Authors,

I read with interest the manuscript. However, the presence of relevant methodological limitations has strongly limited the study. 

The main weakness of the study regards the participants recruitment, especially the inclusion/exclusion criteria that has been adopted.

In general, your Participants subsection (2.1) of the Methods substantially summarized information that should instead be included in the Results (I kindly suggest moving to Results the entire information contained in the Participants subsection). The Participants subsection should rather include a description of the setting (which hospital, which ward), a description of the recruitment procedure (who carried out the assessments, inclusion/exclusion criteria for participating).

However, the very limiting point is related to the lack of a clear description of the inclusion/exclusion criteria, which is source of relevant methodological issues. For instance, it seems that you did not account for history of psychiatric and/or neurocognitive disorders; nor you evaluated the presence of current psychiatric and/or neurocognitive disorders that could have biased the interviews. This is just an example of the degree of bias in the selection of participants of the present study, as well as in the interpretation of findings.

I strongly recommend the Authors to clearly describe inclusion/exclusion criteria, even before I could debate on the Results. 

I recommend to submit a new version of the manuscript, after overcoming this relevant limitation. 

Author Response

Thank you very much for your comments to improve the quality of this articles.

The revised parts were marked in red, we included the page and line of the revised part.

Response to Reviewer 3 Comments

Point 1. The main weakness of the study regards the participants recruitment, especially the inclusion/exclusion criteria that has been adopted.

Response 1: Thank you very much for your good comment. We had a criteria for selecting participants, but we made a mistake not to include them in the manuscript. So, we added the criteria for inclusion/exclusion of participants as below.

“Criteria for inclusion of participants are as follows. The first was an inpatient aged 18 or older who have been diagnosed with alcohol use disorder based on the DSM-5 criteria [19]. The second was the alcoholic patient is able to understand and respond to items of questionnaires and communicate. To meet the second criterion, alcoholic patients with intellectual disability, mild cognitive impairment as well as dementia, and schizophrenia spectrum disorder such delusional disorder, schizophrenia, and schizoaffective disorder as a co-occurring disorder were excluded.” (p. 3, L 8-13).

Point 2. In general, your Participants subsection (2.1) of the Methods substantially summarized information that should instead be included in the Results (I kindly suggest moving to Results the entire information contained in the Participants subsection).

Response 2: As you advised, we added the characteristics of the participants to the result section. We transfer some from methodology section to result section, and added some contents and a table including patients’ characteristics such as age of first alcohol use, age at which regular drinking began, age of diagnosed with alcohol use disorder, hospitalization periods for the past year, and presence of COD. (p. 5, L 5-20).

Point 3. The Participants subsection should rather include a description of the setting (which hospital, which ward), a description of the recruitment procedure).

Response 3: Thank you for your advice. As you advised, in addition to the criteria for inclusion/exclusion of participants, we also added information about hospitals and wards as follows.

“The participants were 101 male and female alcoholic patients diagnosed with alcohol use disorder and hospitalized in Chamsarang Hospital, which specializes alcohol treatment at Incheon, Korea. The patients who participated in the study were hospitalized in Ward 3, 4 and 5. Ward 3 was for patients only with alcohol use disorder, Ward 4 was for male patients with alcohol use disorder and mental disorders as co-occurring disorders (COD), and Ward 5 was for female patients with both alcohol use disorder and mental disorders.” (p. 3, L 4-7).

Round 2

Reviewer 2 Report

Authors failed to address my concerns. In my opinion, the paper does not have the quality to be published in a Q1/Q2 journal in terms of novelty and research quality.

Author Response

Response to Reviewer 2 Comments

Point 1. Authors failed to address my concerns. In my opinion, the paper does not have the quality to be published in a Q1/Q2 journal in terms of novelty and research quality

Response 1: I agree, such criteria for multicollinearity, a tolerance must be more than .1 and VIF must be less than 10 are too permissible for researchers. We believe that, more conservatively, all tolerance was more than .4 and VIFs were less than 3 indicates that there is no significant problem with the multicollinearity among predictors included in this study.

Of course, no regression model could be completely free from multicollinearity. We revised the description of the examination of the multicollinearity and also presented the Durbin Watson statistics as follows.

“It is known that statistically multicollinearity problem occurs when a tolerance is less than .2 or .1, and a variance inflation factors (VIF) is greater than 5 or 10 [30]. Because tolerances of predictors in this study were .442 ~ .689, and VIFs were 1.451 ~ 2.265, the multicollinearity problem was not significant. And, a value of the Durbin Watson statistic was 2.251 which indicates that there is no autocorrelation detected in the sample because it was close to 2. (p. 6, L 15-18).

We humbly accept comments regarding novelty and research quality of this study, and acknowledge that there are limitations of the study, as stated in the text. Despite of the limitations, we believe that the results of this study conducted with alcoholic inpatients would contribute both academically and clinically.

Reviewer 3 Report

I would sincerely thank the Authors for having welcomed my suggestions. The Participants section appears now improved.  

Author Response

Point 1. I would sincerely thank the Authors for having welcomed my suggestions. The Participants section appears now improved.

Response 1: Thank you so much for your good evaluation. We will continue to make efforts to improve the quality of this articles.

The revised parts were marked in red.

We asked editing company to correct it in English before first submission. If the final decision is to be approved for publishing this manuscript, I will request that editing company to correct English sentences and grammar once again.
